# Tourism Competitiveness Evaluation Model of Urban Historical and Cultural Districts Based on Multi-Source Data and the AHP Method: A Case Study in Suzhou Ancient City

**Yao Lu, Mao-en He and Chang Liu \***

College of Design and Innovation, Tongji University, Shanghai 200092, China; 2133642@tongji.edu.cn (Y.L.);
hemaoen@tongji.edu.cn (M.-e.H.)
* Correspondence: liuc@tongji.edu.cn

**Abstract:** Urban historical and cultural districts, serving as multi-functional compounds integrating cultural preservation, consumer experience, and economic growth, are increasingly becoming the preferred choice for in-depth tourism under the trend of historical heritage protection and consumption upgrading. Due to the complexity of the construction purpose, inherent functions, and operational management of historical districts, scientifically and rationally evaluating them poses a challenge. This paper attempts to construct an evaluation method for the tourism competitiveness of urban historical and cultural districts based on multi-source data and the Analytic Hierarchy Process (AHP) method. First, based on the model of destination competitiveness and combined with literature research and open-ended expert interviews, an evaluation framework for the tourism competitiveness of urban historical and cultural districts is established, using the AHP method to calculate the specific weights of each evaluation indicator. Then, the corresponding data sources for each indicator and the data processing and calculation methods are further clarified. To verify the effectiveness of the proposed evaluation model, this paper selects three key historical and cultural districts in Suzhou City, calculates the tourism competitiveness of each district based on the proposed model, and collects tourist satisfaction surveys from the three districts for cross-validation with the evaluation results. The experimental results show that the evaluation model is reliably effective in assessing the cultural, commercial, and tourism service aspects of historical districts, thereby providing a theoretical basis for future tourism decision-making information systems and practical applications of historical districts.

**Keywords:** tourism competitiveness evaluation; urban historical and cultural district; multi-source data; AHP method

## 1. Introduction

Urban historical and cultural districts, as significant carriers of the tourism economy, have garnered considerable attention from tourism economy managers and practitioners regarding their planning, construction, and operation. In China, with the improvement of people's living standards, there is a growing preference for tourism products with cultural connotations, and the number of tourists visiting cultural heritage sites is showing an explosive growth trend [1]. Currently, a large number of historical and cultural district renovation projects have emerged in various cities. However, as the transformation and operation of these districts often adopt a top-down model led by local governments blindly copying the business models of other cities' districts, many cities in China face issues such as a lack of distinctiveness, vitality, severe homogenization, and weak sustainability in their historical and cultural districts. This has led to aesthetic fatigue among tourists for such cultural districts and, subsequently, to a serious decline in visitor numbers and economic benefits in these cities' historical districts after initial market prosperity.

The above issues can essentially be attributed to a lack of tourism competitiveness in historical districts. The European Union officially defines competitiveness as an indicator

for measuring sustained productivity, with the level of competitiveness determining to what extent competitors can effectively promote income and welfare growth [2], ultimately deciding whether competitors will succeed or fail [3]. As a tourism destination, historical and cultural districts must continually enhance their market competitiveness to achieve sustainable development. Therefore, an objective evaluation of the tourism competitiveness of historical and cultural districts will help improve their competitiveness in the tourism market for sustainable development. On the one hand, in terms of research on competitiveness evaluation systems, related tourism competitiveness evaluations mainly focus on the destination's resources, local cultural preservation, and landscape creation from the perspectives of tourism studies and environmental or urban planning, or they explore the competitiveness of tourism economics and shopping experiences from a business operation perspective. However, urban historical and cultural districts, as entities with both cultural tourism and commercial attributes, require a comprehensive evaluation of their competitiveness from cultural resources, tourism resources, and commercial resources. On the other hand, in terms of the collection and calculation of quantitative data, traditional research methods mainly use quantitative questionnaires to collect data and employ multi-criteria decision analysis methods (such as SMAA, PROMETHEE, AHP, etc.) to measure the final results. However, due to the dynamic nature of competitiveness, which requires the ability to quickly integrate, reconfigure, acquire, and release attractive resources to adapt to the constantly changing market [4], traditional methods lack timeliness and, thus, may not meet the needs for real-time dynamic assessment of historical district competitiveness.

This paper aims to establish an evaluation system for the tourism competitiveness of urban historical and cultural districts. First, based on relevant theories of tourism competitiveness and combined with open-ended expert interviews, an evaluation framework for the tourism competitiveness of urban historical and cultural districts is proposed and concretized into 29 indicators, including data sources. Then, the Analytic Hierarchy Process (AHP) is used to calculate the weights of each indicator, completing the construction of the evaluation system for the tourism competitiveness of cultural districts. To verify the effectiveness of the proposed evaluation system, Pingjian, Shantan, and Guanqian (three important historical districts in Suzhou) are selected for evaluation, with competitiveness scores calculated based on the proposed system and compared with field research and subjective user evaluations.

The contributions of this paper can be summarized in three aspects. Firstly, in terms of theoretical research, the proposed evaluation system for the tourism competitiveness of historical and cultural districts offers a comprehensive assessment of the cultural, commercial, and tourism attributes of historic districts. This approach addresses the issue of existing systems that focus solely on either cultural preservation assessment or tourism economic evaluation, lacking a comprehensive framework. It aligns with the policy needs for the integrated development of culture and tourism in historic districts in the new era. Secondly, regarding the assessment methodology, this paper introduces a method for calculating the competitiveness indicators of historic districts based on multi-source objective data. This method enhances the timeliness and reliability of the calculations. Thirdly, from the perspective of practical industry and application, the proposed evaluation system assists managers of historic districts in exploring balanced development models between cultural preservation and commercial development in tourism destinations. It enables them to specifically coordinate and arrange resources in historic districts, adjust and optimize operational models, and address current issues in historic districts related to insufficient protection and utilization of cultural and tourism resources. On the other hand, the calculation method based on multi-source objective data facilitates real-time and dynamic decision-making, ensuring that managers can promptly understand consumers' dynamic needs. This, in turn, serves as an effective reference for improving the quality of tourism services and activating underestimated potential tourism highlights [5].

## 2. Related Work

### 2.1. Urban Historical and Cultural Districts

Urban historical and cultural districts refer to historical and cultural preservation areas located in cities that reflect traditional characteristics [1]. The concept of cultural and historical districts can be traced back to the Athens Charter of 1931, which mentioned the importance of protecting the areas surrounding historical sites. The term "Historic Areas" was used in the 1933 Athens Charter, encompassing the sites of protected ancient buildings and their surrounding environments in the overall consideration of historical preservation. In 1987, the Washington Charter introduced "Historic Urban Areas", clearly establishing the foundational concept of historical and cultural districts as both material and spiritual entities, comprising an organic whole of their form, scale, architecture, surrounding environment, functions, history, and cultural crafts. In China, historical and cultural districts are defined as areas designated and published by provincial, autonomous region, or municipal governments that are rich in cultural relics, have a concentration of historical buildings, can reflect traditional patterns and historical styles in a complete and authentic manner, and have a certain scale, thus reflecting their administrative attributes. Functionally, historical and cultural districts are an important part of China's cultural heritage and also key carriers of cultural heritage tourism, serving as destinations for tourists seeking to experience local culture, art, historical architecture, traditional life, and artifacts [6].

The functional attributes of urban historical and cultural districts have evolved over the past decades, gaining many new characteristics. Prior to the 1990s, the perception of historical and cultural districts was mainly influenced by the "curatorial approach", which saw these areas as cultural heritage to be protected in their original state, with financial conditions and public needs in these areas being considered secondary, and some heritage managers even viewed the arrival of tourists as a disturbance to the cultural heritage itself [7]. However, with the development of sustainable tourism concepts, both managers and the public have come to realize that the economic status of urban historical and cultural districts is an integral part of sustainable tourism and conducting certain commercial activities within these districts is an important means for tourism destinations to obtain funds to support sustainable tourism.

Today, commerce is an indispensable component of urban historical and cultural districts, not only because the tourism industry plays an increasingly important role in the current economic system but also because cultural tourism experiences have become more diverse and wide-ranging. These experiences rely on the rich output of local cultural industries and the provision of excellent "Tourism-Specific Products" to enhance tourists' travel, cultural, and entertainment experiences [8]. Moreover, as tourism destinations within the city, historical and cultural districts are fully integrated into the city's tourism system. The functional attributes of urban historical and cultural districts have evolved from the singular attribute of cultural sites to destinations with diversified functions encompassing culture, commerce, and tourism.

In summary, the tourism experience of urban historical and cultural districts is influenced by a variety of factors, including local cultural heritage, cultural industries, consumer commerce, and tourism services.

### 2.2. Competitiveness of Tourism Destinations

Tourism destinations are increasingly regarded by researchers as "space tourism service providers offering unique attractions or features" [9]. As the role of these destinations as "providers of tourism experiences" gains importance, the competitiveness of tourism destinations has come into focus. The competitiveness of a tourism destination is considered a measure of the degree of success achieved by the destination and the ability of the destination, as a whole, to offer tourist experiences that surpass those of potential competitors [10].

In the early 1980s, the competitiveness of tourism destinations mainly included two key elements: profitability and sustainability. Buhalis [11] summarized it as the "long-term



success of a tourism area". The study by Crouch and Ritchie [10] is a significant reference in this context. Their research on the competitiveness of tourism destinations is based on Porter's "National Competitiveness Diamond" model (including factor conditions, demand conditions, related and supporting industries, firm strategy, structure and rivalry, chance, and government [12]). According to Crouch and Ritchie, for a tourism destination to achieve economic success, it must pay attention to environmental management, infrastructure, quality of life, and internal industries. This is a systematic process that examines a destination's ability to manage and organize internal resources driven by competitive strategies based on theories of comparative advantage, competitive advantage, and the characteristics of the micro and macro environments in which the competitive tourism destinations operate. With the continuous development of the concept of tourism destination competitiveness, academic institutions and organizations have also conducted competitiveness studies for different types of tourism destinations. One of the most representative research outcomes is the Travel and Tourism Competitiveness Index published by the World Economic Forum (WEF, https://www.weforum.org (accessed on 24 September 2023)) since 2007, which comprehensively examines factors such as national socio-economic conditions, healthcare, working conditions, and air transport infrastructure at the national level, making it one of the most influential report series on international tourism destination competitiveness.

For small-scale research subjects like urban historical and cultural districts, their competitiveness must be defined in full consideration of the characteristics. On one hand, urban tourism, as a widespread and rapidly growing phenomenon [13], is shaped by the basic conditions of the city, including population, industrial structure, economic development level, healthcare conditions, geographical location, infrastructure, and policy-making, which constitute the external environment of the historical and cultural districts. On the other hand, as cultural tourism destinations, the tourist experience becomes a priority in local tourism management [6]. The tourist experience in historical and cultural districts revolves not only around tangible sites, monuments, and landscapes but also relates to local lifestyles, creative products, and "everyday culture" [14]. Excellent tourist experiences, destination image, and tourist engagement are key reasons for enhancing tourist satisfaction [15–19], and they collectively form the brand perception of a tourism destination. All products and services are constrained by this unified brand perception [9], implying that for tourism destinations, the tourist experiences generated through competitive actions can be translated into tourist satisfaction and happiness. Competitiveness assessments based on satisfaction surveys can, to some extent, validate research on competitiveness assessment from a "supply perspective", further enhancing the scientific nature of the competitiveness analysis model.

### 2.3. Multi-Source Tourism Data and Analytic Hierarchy Process (AHP) Method

Continuously assessing tourism destinations is instrumental in enhancing their competitiveness. Vanhove [8] posited that a tourism destination does not gain a competitive advantage by chance but as a result of sustained planning. Research in the business sector shows that enterprises employing Decision Support Systems (DSS) and planning procedures perform better competitively than those that do not [8], a finding that can similarly be applied to the management and operation of tourism destinations. The foundation of scientific decision-making lies in using real data for accurate evaluations, and traditional methods of assessing tourism destinations struggle to collect developmental data from within the districts in a timely, ongoing manner, necessitating innovative approaches in data collection.

Utilizing current, multi-source public data for data collection and analysis of destinations is an increasingly feasible and common approach in recent tourism research. For instance, Liu et al. [20] used geotagged photo and video data from Instagram to analyze the preferences of tourists from different countries for types of tourism destinations in the Kowloon area of Hong Kong. Li et al. [21] employed street video data combined with deep learning algorithms to analyze the relationship between the environmental form of districts

and their vitality. Walz et al. [22] used the Official Topographical Cartographic Information System of Germany to calculate the landscape attractiveness of different regions in Germany. Data from OpenStreetMap (OSM), such as Points of Interest (POI), have been used in scenarios like analyzing urban traffic characteristics [23], and Huang et al. [24] combined Instagram and Twitter data to study destination images in parts of Poland.

On the other hand, the Analytic Hierarchy Process (AHP) is a well-established multi-criteria analysis method extensively applied in complex research scenarios aimed at reducing the emotional burden in decision-making by comparing different criteria [25]. It transforms experts' subjective opinions into high-quality decisions and can be applied to any criterion-based decision-making scenario [26], mainly addressing selection/evaluation problems in tourism research. For the application of AHP in competitiveness assessment, Zhou et al. [27] used a mixed AHP approach based on the perspective of destination resources to assess the tourism competitiveness of West Virginia, and more applications of AHP are seen in various tourism analyses, including but not limited to tourism destination ranking, tourism strategy formulation, tourism attraction evaluation, and personalized attraction recommendations [28–30], making it one of the most common methods of analysis and evaluation in tourism studies.

However, it is undeniable that the AHP method involves experts' preferences from its inception, inevitably leading to subjective biases in the assessment results [31]. This presents two requirements for constructing evaluation systems: first, the selection of the expert group, that is, whether the interviewed experts can discuss and judge issues from a relatively objective standpoint; second, the verification of results, namely whether the district competitiveness evaluation obtained through the AHP method can truly match tourists' satisfaction with their experience at the destination and discuss strategies for correcting any initial biases.

## 3. Framework for Tourism Competitiveness of Urban Historical and Cultural Districts

### 3.1. Supply-Driven Evaluation Perspective Based on Crouch and Ritchie's Destination Competitiveness Framework

In constructing the tourism competitiveness evaluation system for urban historical and cultural districts, the study primarily references the destination competitiveness model of Crouch and Ritchie. According to their model, the competitiveness of a tourism destination is divided into four fundamental dimensions: Core Resources and Attractors, Supporting Factors and Resources, Destination Management, and Qualifying Determinants. However, based on the purpose of the study and the characteristics of urban historical and cultural districts, we have categorized the basic dimensions of their competitiveness as Core Resources and Attractors, Supporting Factors and Facilities, and Guarantee Factors.

1.  Core Resources and Attractors follow Crouch and Ritchie's definition, representing the main components of a destination's attractiveness and the primary motivations for a tourist to visit a destination. It is the presence of these attractors that leads tourists to choose a particular destination for their travel activities.

2.  Supporting Resources and Facilities are adapted from Supporting Factors and Resources. In the original definition, this dimension represented the solid foundation for tourists to engage in tourism activities at the destination. Supporting Resources and Facilities are mainly divided into two aspects: "hardware", primarily reflected in tourism and basic infrastructure, and "software", reflected in the local commercial atmosphere and quality of public services.

3.  Guarantee Factors are evolved from Qualifying Determinants. In the original definition, Qualifying Determinants can be understood as "contextual conditions" that determine the scale, restrictions, and potential of a tourism destination, including its geographical location, competitive environment, safety, and cost. However, for urban cultural districts, their geographical location and competitive factors are more reflected in the geographical situation of the cities they are located in. Therefore, this

study mainly considers safety and cost factors, viewing them as a type of tourism guarantee provided by the locale.

It should be noted that the objective of this study is to assist local tourism managers in improving their tourism management level and better utilizing internal and external tourism resources of the district to enhance their competitiveness. Therefore, content related to Destination Management has been omitted in the assessment dimensions.

### 3.2. Indicator Selection Based on Desk Research

In the dimensional hierarchy following the basic dimensions, dimensions were determined by referencing the tourism evaluation literature and synthesizing expert group opinions. Since the referenced tourism competitiveness model included elements related to attractions for urban historical and cultural districts, which are a composite of cultural, commercial, and tourism destinations, it was appropriate to refer to both the tourism and commercial attractiveness assessment literature. Table 1 lists some of the evaluation dimensions frequently mentioned in the literature, which will be incorporated and modified during the expert discussion process.

**Table 1.** Various Evaluation Dimensions Mentioned in the Literature.

| Dimensions/Scholars * | A [32] | C and A [6] | C and N [9] | C and R [10] | G [33] | H [34] | L and B [19] | M [35] | P [36] | R and Z [37] | T and R [38] | Y and R [39] | Total |
|---|---|---|---|---|---|---|---|---|---|---|---|---|---|
| Climate | | * | | * | * | | | | * | * | | * | 6 |
| Social Culture | * | * | * | * | * | | | | * | * | | | 7 |
| Historical Assets | * | * | | | * | * | | | * | * | | * | 7 |
| Traditional Crafts | | * | | | * | | | | | * | | | 3 |
| Infrastructure | * | | | * | * | | | | | * | * | * | 6 |
| Safety | | * | * | * | | * | | | * | | | | 5 |
| Price | | * | | * | | | | | * | * | | * | 5 |
| Events and Activities | | * | | * | * | | * | | * | | | * | 6 |
| Commercial Quality | | * | | * | * | | | | * | * | * | * | 7 |
| Accessibility | * | | | * | * | * | | | * | * | * | * | 8 |
| Accommodation Facilities | | * | | | * | * | | | * | * | | * | 6 |
| Local Food | | | * | | * | | | | * | * | | * | 5 |
| Religion | | * | | | * | | | | | * | | | 3 |
| Information Services | * | | * | * | * | | | | * | | | | 5 |
| Recreation and Entertainment | | | | | * | * | | | * | * | | * | 5 |
| Resident Friendliness | * | * | * | | | | | | * | * | | * | 6 |
| Reputation | | * | | | | * | | | * | | | | 3 |
| Environmental Sanitation | | * | | | | * | | * | | | | | 3 |

* Use initials to represent the scholar's name.

As shown in Table 1, Accessibility, Social Culture, Historical Assets, and Commercial Quality are the most frequently mentioned factors (seven times or more), reflecting their significant role in the evaluation of urban historical districts. These indicators should also be considered in the subsequent construction of the evaluation system. Following these, factors such as Climate, Infrastructure, Accommodation Facilities, Resident Friendliness, Safety, Price, Local Food, Information Services, and Recreation and Entertainment are mentioned next (five times or more). Traditional Crafts, Religion, and Environmental Sanitation are mentioned the least (no less than three times), necessitating further analysis to determine their necessity for inclusion within the evaluation system.

*3.3. Indicator Selection Based on Expert Interviews*

In the open-ended interview phase, this study set up a group of six experts to discuss the evaluation framework based on existing desk research and perform AHP weighting. The setup of the expert group followed these principles:

1. Experts needed to have a knowledge background that reflects a thorough understanding of historical and cultural districts as well as tourism.
2. The expert group could evaluate the competitiveness of district tourism from the perspective of different stakeholders in tourism behavior.

Specifically, to ensure the representativeness of each expert, we set the following criteria during the selection process for each category of experts:

1. Expertise Area: Select representative experts from three groups: theoretical researchers of historical districts, managers and operators of historical districts, and experiencers of historical districts.
2. Qualifications Review: Experts participating in the interviews should possess high-level professional capabilities and many years of work experience (at least 10 years of relevant field research, practice, or experience). To ensure the knowledge level of experts, the minimum educational qualification for managers and visitor-type experts is set at a bachelor's degree or above, while theoretical research experts are required to have a doctoral degree. In terms of professional titles, priority is given to heads and deputy heads in related fields from both public and private sectors, as well as to associate professors and higher.

Therefore, this study invited three types of experts (two from each category): the first category consisted of managers of historical and cultural districts and internal heritage conservation units (DM, District Manager), the second category included seasoned travelers and tourism experience experts (TEE, Tourism Experience Expert), and the third category comprised scholars with a background in tourism architecture, urban planning, and environmental design (AE, Academic Expert). Table 2 specifically details the types, positions, and backgrounds of these experts.

**Table 2.** Detailed Introduction of Experts.

| Expert Type | Position | Experience |
|---|---|---|
| DM | Director of Humble Administrator's Garden, curator of Suzhou Garden Museum | Holds a bachelor's degree with over 20 years of experience in the relevant field. Manages the Humble Administrator's Garden, a representative of the classical gardens in the Jiangnan region and one of China's most important cultural heritages. The garden was listed as a World Heritage Site by UNESCO in 1997. |
| DM | Head of the Suzhou Tourism Bureau, deputy director of Suzhou Cultural and Creative Industry Development Center | Holds a bachelor's degree and has over 20 years of experience in the relevant field. A key expert-type manager in the development direction of Suzhou's cultural tourism and cultural creative industries. |
| TEE | Head of the tourism experience department of Tongcheng Travel | Holds a master's degree with over 15 years of experience in the relevant field. Currently employed at Tongcheng Travel, established in 2004, which is the second-largest internet tourism company in China. |
| TEE | Senior sojourner | Holds a bachelor's degree and is a seasoned sojourning expert with over 10 years of sojourning experience, having traveled extensively across China. |

**Table 2.** *Cont.*

| Expert Type | Position | Experience |
|---|---|---|
| AE | Professor from Tongji University specializing in urban planning and environmental design | Holds a doctoral degree with over 20 years of research experience in the professional field. Served as the secretary of the 7th Discipline Evaluation Group (Landscape Architecture Group) of the State Council Academic Degrees Committee. Special editor for "Chinese Landscape Architecture" and has published more than 40 papers in core journals, including SSCI and CSCI. |
| AE | Senior researcher from the School of Design and Innovation specializing in urban planning and environmental design | Holds a Doctoral degree and is a serial entrepreneur with over 15 years of research experience in the professional field. Registered as a planner in the Netherlands and serves as a senior architect at AECOM. Recipient of multiple international awards and has led the planning and design of significant projects, including the Shanghai Bund Art Museum. |

Each expert discussed the dimensions, hierarchy, and rationality of each dimension of this competitiveness model through open-ended interviews guided by the following questions:

- Do you think the classification of these competitiveness evaluation dimensions is reasonable? Are all relevant aspects covered? Are the dimensions independent of each other?
- Do you think the wording of these dimensions is appropriate and fully reflects the core characteristics of historical and cultural districts?
- How do you define the competitiveness of urban historical and cultural districts? What does competitiveness mean for a district?
- Can these dimensions provide guidance for improving the management of tourism destinations during the evaluation process?

Based on the desk research summary and the open-ended interviews with the expert group, the final compilation formed the urban historical and cultural district competitiveness evaluation framework, as shown in Figure 1.

It should be noted that the model proposed by Crouch and Ritchie is generally considered a theoretical framework and does not directly yield specific indicators. Instead, it requires adjustments and refinement based on the specific characteristics of the research or application subject. Specifically, in constructing a tourism competitiveness evaluation system for urban historical and cultural districts, we reconsidered unique attractors of such districts (e.g., Cultural Specialty Consumption Places) under the dimension of "Core Resources and Attractors". These types of commercial units are a vital part of the tourism experience in historical and cultural districts, reflecting the integration of culture and commerce. The dimension of "Supporting Factors and Facilities" largely follows the original analytical framework's definition but transforms the more abstract concept of "resources" into "facilities" specific to historical districts. The most significant adjustments were made in the "Guarantee Factors" dimension. In the original definition of "Qualifying Determinants", this dimension required considering factors like the geographical location of the tourism destination, competitive environment, cost, and safety. However, aspects like geographical location and competitive environment require complex evaluation mechanisms and tend more towards qualitative analysis. Moreover, these factors cannot be changed solely by the actions of district managers. Therefore, we streamlined the indicators for this dimension, retaining factors that reflect tourism safeguard measures such as cost and safety.

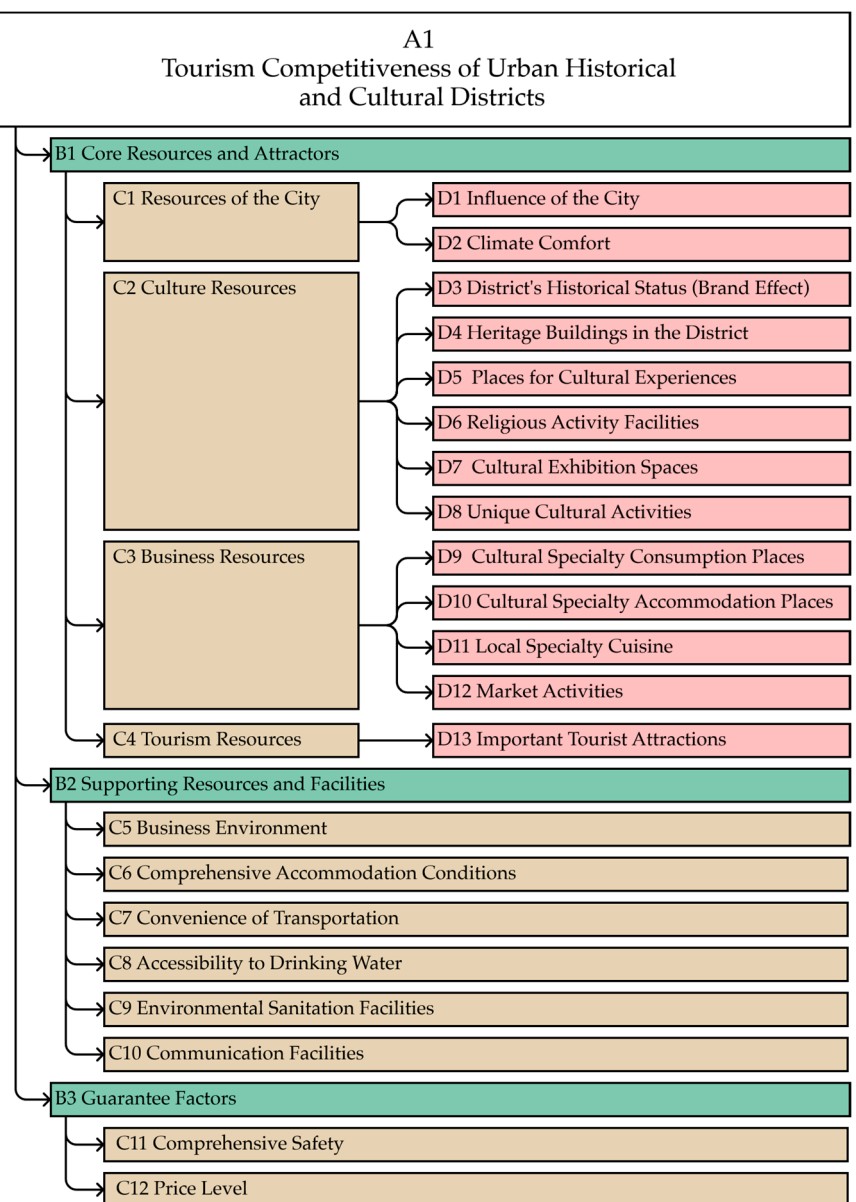

**Figure 1.** Urban Historical and Cultural Districts Tourism Competitiveness Evaluation Framework.

In this framework, A1 is the final indicator for evaluating district competitiveness, while the B-class indicators correspond to Crouch and Ritchie's competitiveness analysis framework, examining the district's core attractiveness resources for tourists, the support capability for tourism activities, and tourist guarantees. In the more basic C-class (execution layer) indicators, C1–C4 as indicators measuring the core resources of the district consider both external environment (C1) and internal resources (C2–C4) of the district; C5–C10 consider both "software facilities" (C5) and "hardware facilities" (C6–C10) supporting destination tourism activities; C11–C12 cover the two guarantee factors most concerned by tourists, namely safety and price.

Specifically, C1, representing "Resources of the City", indicates the comprehensive characteristics of the city where the historical district is located. This includes differences in economic development levels, geographical positions, and climatic conditions of various cities, which, as external competitive environments, significantly affect the tourism competitiveness of their historical districts. C1 is determined by two sub-indicators: D1 ("Influence of the City", reflecting the status and reputation of the city where the district is

located) and D2 ("Climate Comfort", assessing the district's climate suitability for tourism and sojourning activities).

C2, "Culture Resources", reflects the richness of various historical buildings, cultural facilities, and cultural activities within the historical district. It is represented by two types of sub-indicators. The first type reflects the cultural "soft power" (destination marketing effectiveness and local cultural characteristics) through D3 ("District's Historical Status (Brand Effect)"), indicating the importance and marketing reputation of the historical cultural district) and D8 ("Unique Cultural Activities", measuring whether the district possesses distinctive cultural activity resources). The second type reflects the quantity and richness of cultural facilities through D4 ("Heritage Buildings in the District", indicating the richness of cultural heritage buildings in the district), D5 ("Places for Cultural Experiences", assessing the level of cultural experiences in the district), D6 ("Religious Activity Facilities", reflecting the religious activities in the district), and D7 ("Cultural Exhibition Spaces", indicating resources for exhibitions and performances in the district).

C3, "Business Resources", represents the development status of different commercial carriers related to the tourism economy of historical districts. It specifically includes shopping facilities D9 ("Cultural Specialty Consumption Places"), accommodation facilities D10 ("Cultural Specialty Accommodation Places"), dining facilities D11 ("Local Specialty Cuisine"), and market activities D12 ("Market Activities").

C4, "Tourism Resources", measures the richness of tourism resources within the district and is represented by D13 ("Important Tourist Attractions", indicating the level and richness of tourist resources within the district).

To ensure the reliability and consistency of the experts' evaluation results, a rigorous open-ended interview process was established, and specific documentation on the evaluation system and each specific indicator was drafted. Initially, each expert was introduced to the purpose of the evaluation, the theoretical basis and design principles of the evaluation system, and the specific definitions and data sources of each indicator over an hour. This was followed by a discussion with each expert regarding the names, meanings, and specific definitions at the data level of different indicators, reaching a consensus in about 45 min. Subsequently, experts were invited to fill out a pre-designed AHP evaluation questionnaire. Finally, after collecting the questionnaires, we conducted a consistency check for each expert's responses. For experts with many answers violating the transitivity axiom, further communication and confirmation were sought, and the questionnaire results were iteratively refined to ensure reliability and consistency.

*3.4. Data Sources for Indicators*

For the basic execution layer indicators, each requires a corresponding data source and quantification method. After several rounds of expert discussions, we comprehensively considered China's major tourism information platforms and selected AMap, Baidu Map, Dianping, and Ctrip, combined with data from international organizations and Chinese government websites for quantification (as reflected in Figure 2). AMap and Baidu Map are map application platforms in China, functionally similar to Google Maps. However, they possess more extensive POI (Points of Interest) data regarding Chinese geography than Google Maps and offer more detailed information on the delineation of historical districts. Dianping, a subsidiary of Meituan, is a local life information review, sharing, and transaction platform. It is one of China's largest city life consumer guide websites, holding information on nearly all commercial shops and public facilities within Chinese cities (each establishment includes user ratings, reviews, and photos). Ctrip is the largest travel platform in China, accounting for more than half of China's online travel market share. It is also one of the world's largest online travel agencies, encompassing almost all of the scenic areas and hotel data in China (including their geographical location, rating, user

reviews, background information, and more). All data will eventually be linearly scaled to a form between 0 and 1 using the following formula to standardize the range of all data:

$$\text{Final Score} = \text{Original Score}/\text{Max Score} \qquad (1)$$

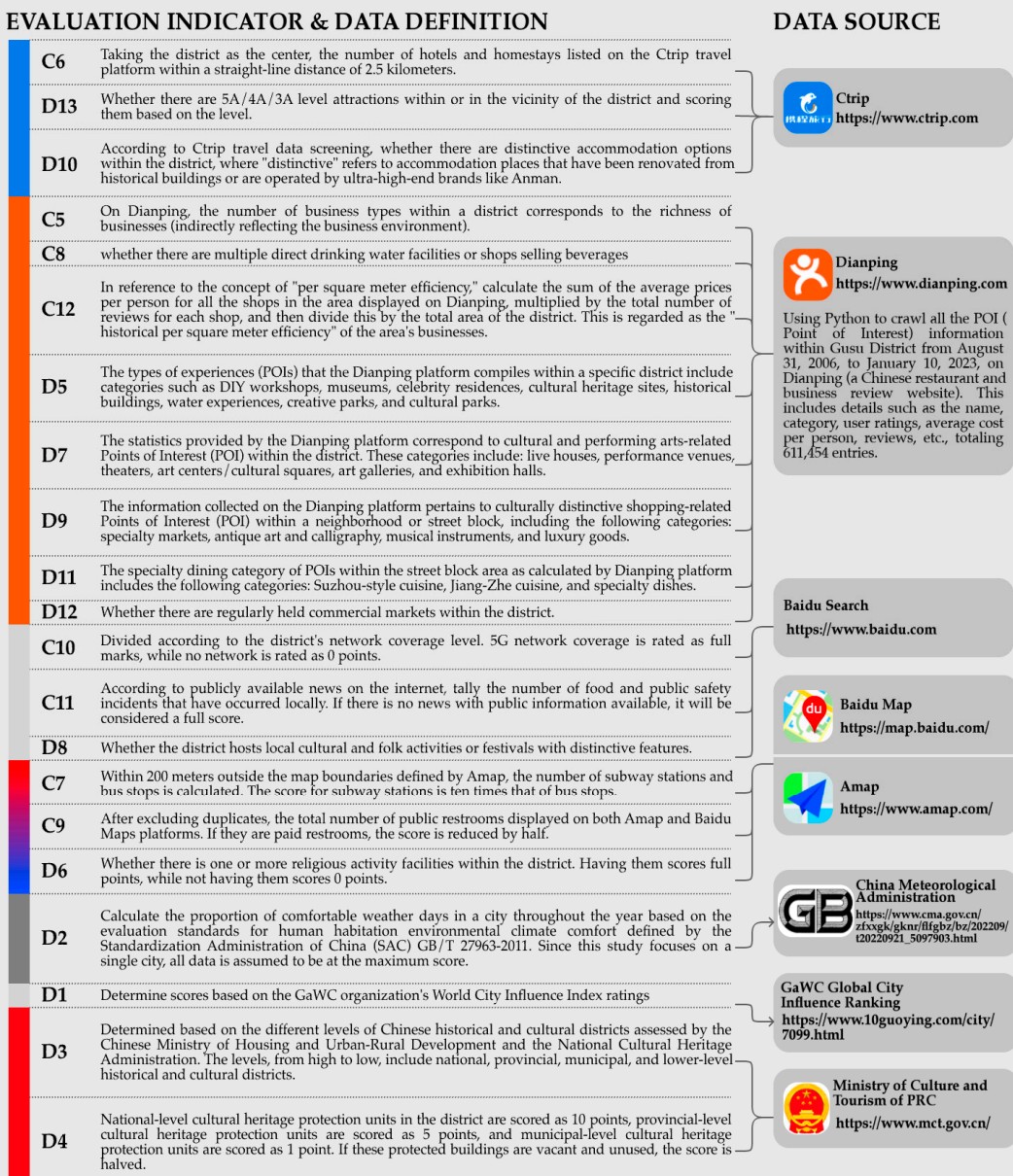

**Figure 2.** Definition and Data Sources of Urban Historical and Cultural District Tourism Competitiveness Indicator Scores.

## 4. Results

### 4.1. AHP Expert Evaluation Questionnaire and Calculation of Indicator Weights

The AHP method assigns weights to indicators by pairwise comparisons within the same level. As shown in Figure 1, all indicators are divided into four levels: A, B, C, and D, encompassing a total of eight categories and 47 questions. The questionnaire uses Saaty's 1–9 scale method to form a 17-level rating scale, which is used to determine the relative importance $a_{ij}$ of indicator i compared to indicator j. Here, a score of 9 indicates that item i is absolutely more important than item j in forming the competitiveness of urban historical

and cultural districts, a score of 1 indicates equal importance, and a score of 1/9 indicates that item i is absolutely less important than item j.

In the calculation, an evaluation matrix for indicators of the same type is first constructed, as shown in Figure 3.

| Indicators | Indicator 1 | Indicator 2 | … | Indicator n |
|---|---|---|---|---|
| Indicator 1 | 1 | $a_{21}$ | $a_{(n-1)1}$ | $a_{n1}$ |
| Indicator 2 | $1/a_{21}$ | 1 | $a_{(n-1)2}$ | $a_{n2}$ |
| … | $1/a_{(n-1)1}$ | $1/a_{(n-1)2}$ | 1 | $a_{n(n-1)}$ |
| Indicator n | $1/a_{n1}$ | $1/a_{n2}$ | $1/a_{n(n-1)}$ | 1 |

**Figure 3.** The Evaluation Matrix.

Next, the eigenvector of the indicator evaluation matrix is solved. The approximate value of each row element's product in the judgment matrix A is calculated using the formula for the nth root of the product:

$$M_i = \sqrt[n]{\prod_{j-1}^{n} a_{ij}} \tag{2}$$

After obtaining the characteristic root of each row element of the matrix, $M_i$ is normalized to obtain the corresponding weight $W_i$ of each indicator:

$$W_i = \frac{M_i}{\sum_{i=1}^{n} M_i} \tag{3}$$

Given that the expert group is divided into three categories, the weights obtained for each indicator by experts of the same type are averaged. The final indicators derived are shown in Table 3.

**Table 3.** Weights of Each Indicator After AHP Calculation.

| Expert Types | Weight of Indicators | | | | | | | | | | | | | | |
|---|---|---|---|---|---|---|---|---|---|---|---|---|---|---|---|
| | B1 | B2 | B3 | C1 | C2 | C3 | C4 | C5 | C6 | C7 | C8 | C9 | C10 | C11 | C12 |
| District Manager | 0.669 | 0.244 | 0.087 | 0.240 | 0.382 | 0.197 | 0.181 | 0.406 | 0.190 | 0.208 | 0.056 | 0.075 | 0.065 | 0.826 | 0.174 |
| Tourism Experience Expert | 0.748 | 0.186 | 0.066 | 0.141 | 0.366 | 0.246 | 0.247 | 0.293 | 0.165 | 0.257 | 0.150 | 0.064 | 0.071 | 0.521 | 0.479 |
| Academic Expert | 0.529 | 0.269 | 0.202 | 0.096 | 0.328 | 0.304 | 0.272 | 0.147 | 0.197 | 0.180 | 0.157 | 0.233 | 0.086 | 0.817 | 0.183 |
| Average | 0.649 | 0.233 | 0.118 | 0.159 | 0.359 | 0.249 | 0.233 | 0.282 | 0.184 | 0.215 | 0.121 | 0.124 | 0.074 | 0.721 | 0.279 |

| Expert Types | Weight of Indicators | | | | | | | | | | | | |
|---|---|---|---|---|---|---|---|---|---|---|---|---|---|
| | D1 | D2 | D3 | D4 | D5 | D6 | D7 | D8 | D9 | D10 | D11 | D12 | D13 |
| District Manager | 0.881 | 0.119 | 0.375 | 0.243 | 0.178 | 0.048 | 0.082 | 0.074 | 0.250 | 0.226 | 0.319 | 0.205 | 1 |
| Tourism Experience Expert | 0.867 | 0.133 | 0.222 | 0.185 | 0.192 | 0.057 | 0.129 | 0.215 | 0.335 | 0.283 | 0.323 | 0.059 | 1 |
| Academic Expert | 0.375 | 0.625 | 0.100 | 0.192 | 0.190 | 0.071 | 0.172 | 0.275 | 0.387 | 0.270 | 0.250 | 0.093 | 1 |
| Average | 0.708 | 0.292 | 0.232 | 0.207 | 0.187 | 0.059 | 0.128 | 0.188 | 0.324 | 0.260 | 0.297 | 0.119 | 1 |

The results of the weight calculation show that for the B-level indicators, experts of all types agree that the B1 Core Resources and Attractors of urban historical and cultural districts have the greatest weight, averaging about two-thirds (0.649) of the components of competitiveness. This is followed by B2 Supporting Factors and Facilities (0.233), and finally, B3 Guarantee Factors (0.118). This indicates a high level of consensus among different types of experts in their understanding of the competitiveness of district tourism. The study also revealed some specific attributes with significantly higher weights, such as D13 Important Attractions in the District (0.151), C11 Safety of Tourist Locations (0.085), C5 Commercial Environment (0.066), D3 Historical Status of the District (Brand Effect) (0.054), D4 Cultural Relics and Buildings in the District (0.048), D5 Experience-oriented Cultural Venues (0.044), etc. Figure 4 shows the evaluation system after the weights have been assigned.

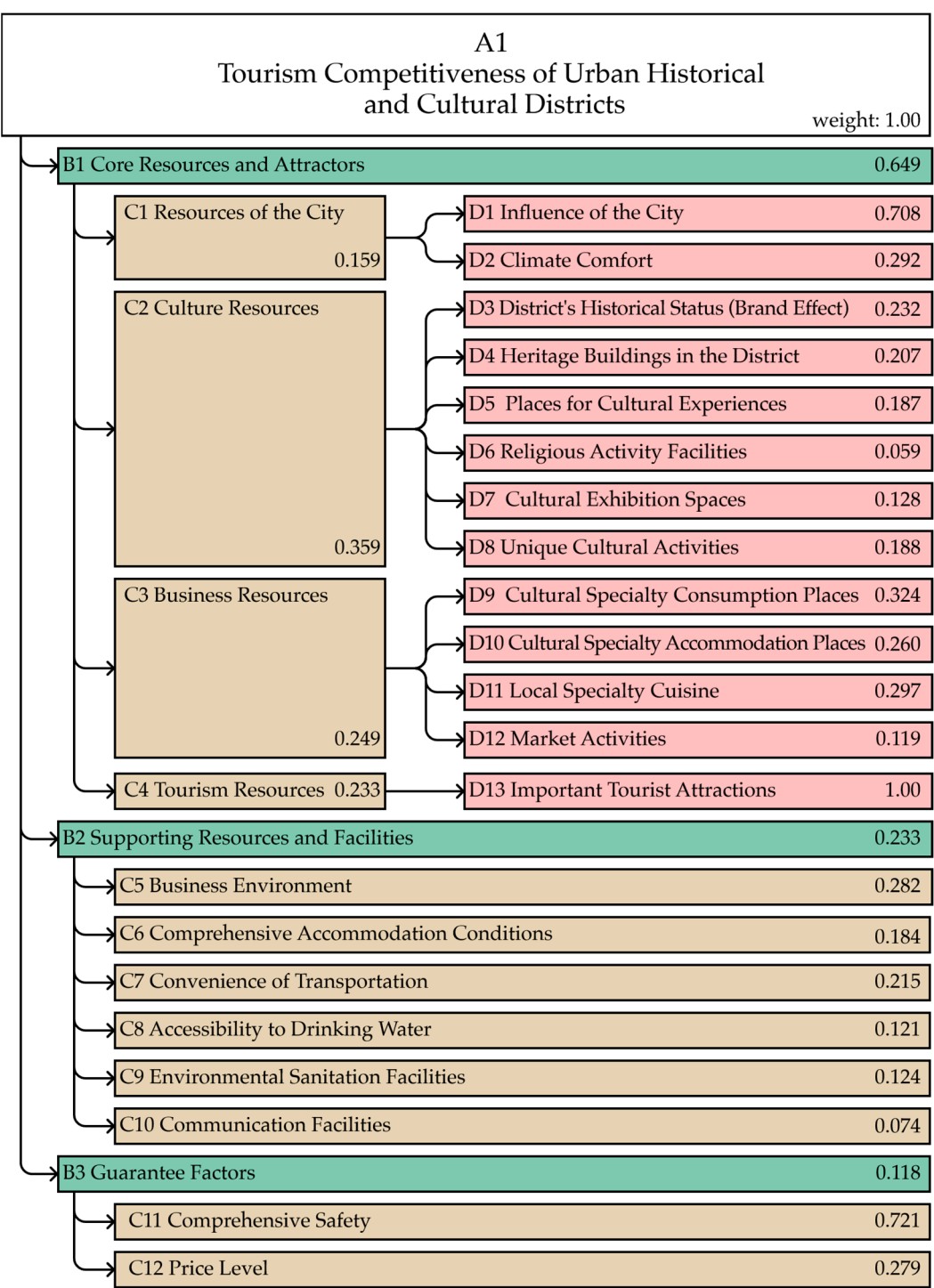

**Figure 4.** Tourism Competitiveness System for Urban Historical and Cultural Districts (Including Weights of Each Indicator).

*4.2. Selection of Suzhou Historical and Cultural Districts as Research Objects and Their Respective Scoring Data*

Suzhou, as a mega-city in China with a core urban population exceeding 5 million, has a historic city center with a history of over 2500 years. This paper selects Shantang (ST), Pingjiang (PJ), and Guanqian (GQ), the three most representative historical and cultural districts within the ancient city of Suzhou, for evaluation. Their forms are shown in Figures 5 and 6.

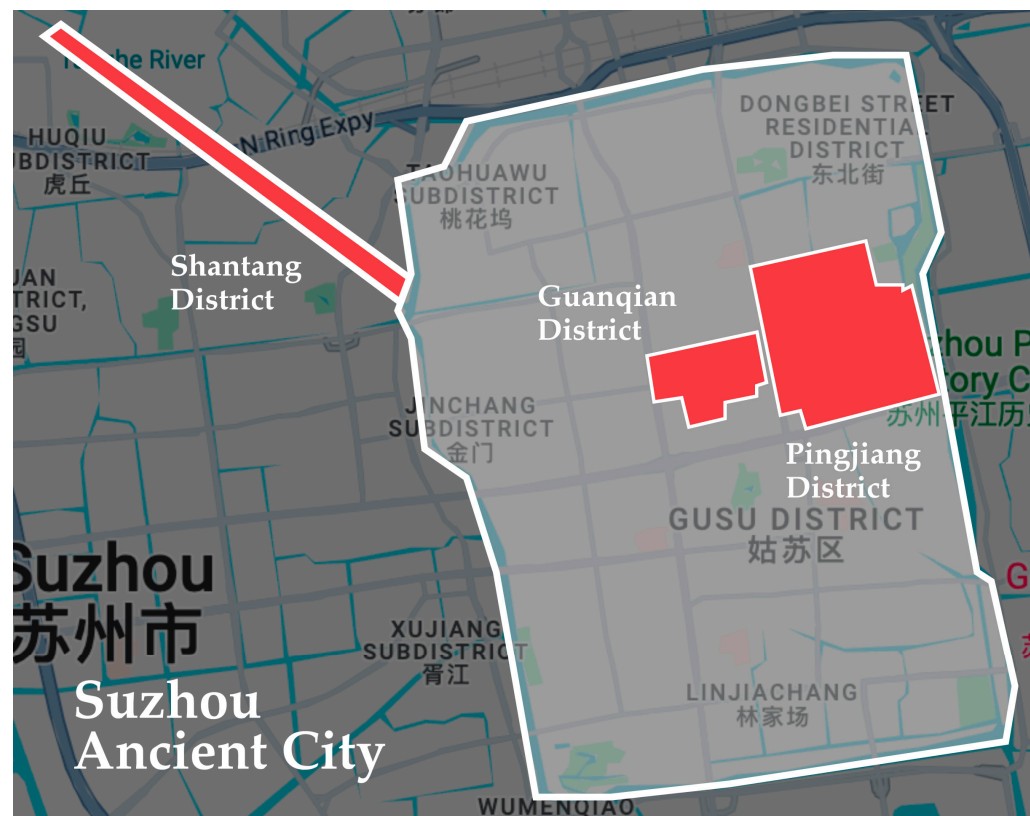

**Figure 5.** The Distribution of Santang, Pingjiang, and Guanqian in the Ancient City of Suzhou.

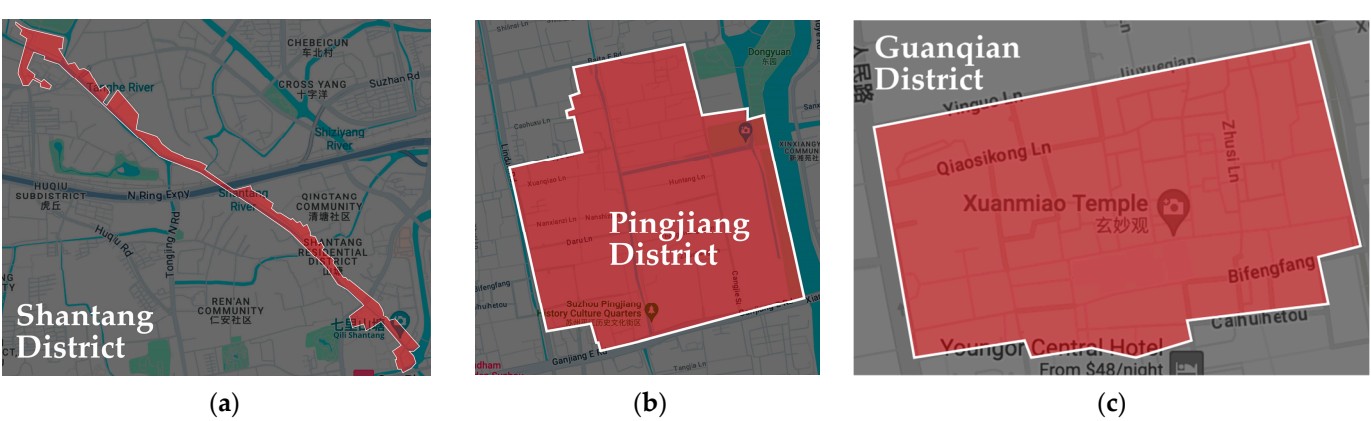

**Figure 6.** (**a**) Shantang District; (**b**) Pingjiang District; (**c**) Guanqian District.

Shantang, established during the Tang Dynasty, was one of the most important commercial districts in ancient China, but its commercial status has declined in modern times. Stretching approximately 3600 m, Shantang houses a multitude of ancient heritage buildings. However, today, less than half of the area has been well-developed as a tourist spot. The other half, containing numerous heritage buildings and sites, remains closed and inaccessible to tourists.

Pingjiang, formed in the Song Dynasty, is the historical and cultural district in Suzhou with the highest quality of preservation and the most comprehensive tourist development. Its cultural heritage richness is similar to Shantang, but it has benefited from better development and utilization. The commercial ecosystem within Pingjiang is also significantly superior to that of Shantang.

Guanqian, dating back to the Song Dynasty, is one of the most representative commercial districts in Suzhou, with its core building, Xuanmiao Temple, constructed during

the Western Jin period. Due to destruction from wars, most of the historical buildings within Guanqian were demolished, and new commercial buildings were erected. Hence, the district now resembles a modern commercial street in its architectural style. Nevertheless, Guanqian still possesses rich cultural content. For example, a multitude of traditional Suzhou delicacies and century-old stores remain active in Guanqian, forming an indispensable part of Suzhou's traditional culture.

Based on the data definition method in Figure 2, we have summarized and organized the specific data for each evaluation indicator of these three historical and cultural districts, which serves as the basis for subsequent score calculations. The data in Table 4 represent the final scores for each indicator. By multiplying the corresponding scores by the previously determined weights, the tourism competitiveness scores for the three urban historical and cultural districts can be calculated and are displayed in Table 5.

**Table 4.** Multi-Source Data Results of Each Indicator.

| Evaluation Indicators | ST | PJ | GQ |
|---|---|---|---|
| | Original Data | Original Data | Original Data |
| C5 | 50 commercial categories | 67 commercial categories | 80 commercial categories |
| C6 | 320 guesthouses and hotels | 216 guesthouses and hotels | 370 guesthouses and hotels |
| C7 | 1 subway station 7 bus stations | 2 subway stations 8 bus stations | 1 subway station 7 bus stations |
| C8 | available | available | available |
| C9 | 20 free public toilets | 17 free public toilets | 14 free public toilets |
| C10 | 5G coverage | 5G coverage | 5G coverage |
| C11 | 0 safety incidents | 0 safety incidents | 0 safety incidents |
| C12 | 21 yuan/sqm | 204 yuan/sqm | 744 yuan/sqm |
| D1 | Gamma+ level | Gamma+ level | Gamma+ level |
| D2 | / | / | / |
| D3 | National level | National level | No official level |
| D4 | 0 national, 3 provincial, 8 city-level, 6 idle | 3 national, 2 provincial, 11 city-level, 6 idle | 1 national, 0 provincial, 2 city-level, 1 idle |
| D5 | 9 venues | 41 venues | 36 venues |
| D6 | Buddhist temple | Buddhist temple | Taoist Temple |
| D7 | 4 spaces | 14 spaces | 18 spaces |
| D8 | None | None | None |
| D9 | 3 venues | 26 venues | 47 venues |
| D10 | None | Available | None |
| D11 | 28 restaurants | 44 restaurants | 27 restaurants |
| D12 | None | Available | Available |
| D13 | 4A level | 4A level | No level |

**Table 5.** Calculated Scores of Each Indicator.

| Evaluation Indicators | ST | PJ | GQ |
|---|---|---|---|
| B1 Core Resources and Attractors | 0.646 | 0.858 | 0.508 |
| B2 Supporting Resources and Facilities | 0.781 | 0.858 | 0.876 |
| B3 Guarantee Factors | 0.729 | 0.797 | 1 |
| C1 Resources of the City | 1 | 1 | 1 |
| C2 Culture Resources | 0.571 | 0.721 | 0.464 |
| C3 Business Resources | 0.198 | 0.835 | 0.736 |

**Table 5.** *Cont.*

| Evaluation Indicators | ST | PJ | GQ |
|---|---|---|---|
| C4 Tourism Resources | 1 | 1 | 0 |
| C5 Business Environment | 0.625 | 0.838 | 1 |
| C6 Comprehensive Accommodation Conditions | 0.865 | 0.584 | 1 |
| C7 Convenience of Transportation | 0.607 | 1 | 0.607 |
| C8 Accessibility to Drinking Water | 1 | 1 | 1 |
| C9 Environmental Sanitation Facilities | 1 | 0.850 | 0.700 |
| C10 Communication Facilities | 1 | 1 | 1 |
| C11 Comprehensive Safety | 1 | 1 | 1 |
| C12 Price Level | 0.028 | 0.274 | 1 |
| D1 Influence of the City | 1 | 1 | 1 |
| D2 Climate Comfort | 1 | 1 | 1 |
| D3 District's Historical Status (Brand Effect) | 1 | 1 | 0 |
| D4 Heritage Buildings in the District | 1 | 0.699 | 0.531 |
| D5 Places for Cultural Experiences | 0.220 | 1 | 0.878 |
| D6 Religious Activity Facilities | 1 | 1 | 1 |
| D7 Cultural Exhibition Spaces | 0.222 | 0.778 | 1 |
| D8 Unique Cultural Activities | 0 | 0 | 0 |
| D9 Cultural Specialty Consumption Places | 0.064 | 0.553 | 1 |
| D10 Cultural Specialty Accommodation Places | 0 | 1 | 0 |
| D11 Local Specialty Cuisine | 0.596 | 0.936 | 1 |
| D12 Market Activities | 0 | 1 | 1 |
| D13 Important Tourist Attractions | 1 | 1 | 0 |

According to the final calculation results in Table 6, regardless of whether for district manager type experts, tourism experience experts, or academic experts, the tourism competitiveness score of Pingjiang is significantly higher than that of Shantang, and the score of Shantang is slightly higher than that of Guanqian. The scoring results of the three types of experts show a certain level of consistency.

**Table 6.** Final Scores of Tourism Competitiveness for The Three Districts.

| Expert Types | Final Score | | |
|---|---|---|---|
| | ST | PJ | GQ |
| District Manager | 0.747 | 0.891 | 0.674 |
| Tourism Experience Expert | 0.654 | 0.840 | 0.591 |
| Academic Expert | 0.694 | 0.841 | 0.690 |
| Average | 0.687 | 0.851 | 0.652 |

*4.3. Comparison and Validation of Evaluation Data*

To further validate the reasonableness of the assessment results, the study surveyed 40 tourists (Among the respondents, one person has an education level below a bachelor's degree, nine have a bachelor's degree, twenty-five hold a master's degree, and five possess a doctoral degree. Additionally, thirteen of the respondents are aged between 20–30, twenty are in the 30–40 age group, and seven are between 40–50 years old. All respondents have visited all three districts: Shantang, Pingjiang, and Guanqian.) using a 5-point Likert scale (where 1 is the worst and 5 is the best) to score their satisfaction with the cultural attractiveness, consumption experience, recreational experience, infrastructure, and overall evaluation of the three districts. The final results are shown in Table 7.

**Table 7.** Results of the Tourist Satisfaction Questionnaire for the Three Districts.

| Evaluation Dimension | ST | PJ | GQ |
|---|---|---|---|
| Cultural Attractiveness | 3.850 | 4.200 | 3.275 |
| Consumption Experience | 3.325 | 3.650 | 3.500 |
| Recreational Experience | 3.700 | 4.100 | 3.400 |
| Infrastructure | 3.850 | 3.925 | 3.925 |
| Overall Evaluation | 3.825 | 4.075 | 3.825 |

From Table 7, it can be observed that by calculating the average scores of four subdivided dimensions, Shantang scores 3.68, Pingjiang 3.97, and Guanqian 3.52, presenting the same ranking result as the AHP calculation. However, in the comprehensive evaluation of the overall integrity of the districts, Guanqian received a score equal to Shantang, resulting in a slight deviation.

## 5. Discussion and Conclusions

### 5.1. Discussion on the Weights of Different Indicators

In this study, B1 Core Resources and Attractors (0.649) accounts for the highest proportion of the tourism competitiveness of urban cultural districts, followed by B2 Supporting Factors and Facilities (0.233), with B3 Guarantee Factors (0.118) having the lowest proportion. This finding aligns with the results of Ritchie and Crouch's study. According to the literature, B1 Core Resources and Attractors is considered the fundamental reason tourists choose one destination over another, forming the most central element of a tourism destination's competitiveness. This consensus is shared by district managers, visitors, and academic experts in the field.

In further discussions about the contributions of different types of attraction resources to the competitiveness of districts, C2 Cultural Resources (0.359) stands out, which is likely related to the characteristics of the study objects. As urban historical and cultural districts are major destination types for cultural heritage tourism, it is predictable that culture itself forms a core component of a district's competitiveness. Notably, the external environment of the district, or C1 City Resources, is assigned a lower weight (0.159). This may imply that even if a historical district is not located in a bustling city, it can significantly enhance its competitiveness and attract a considerable number of tourists by fully utilizing its internal cultural and commercial resources.

There were also certain differences in the weights assigned by different types of experts to some dimensions, as shown in Table 8. Under the factor of cultural resources, both district managers and tourism experience experts assigned the highest weight to D3 District Historical Status (Brand Effect). However, academic experts placed it in a relatively unimportant position. This may be because the historical status of a district is a rating given by relevant institutions, but academic experts are more concerned with the characteristics of the historical district itself rather than third-party evaluations. The academic experts interviewed have a background in urban planning, and they value more the intrinsic qualities that constitute a district's enduring appeal and can support its sustainable development. This reflects that such experts are less influenced by other evaluations and are more inclined to trust their professional judgment, which is why they assigned significantly higher weights to D2 Climate Comfort and C9 Environmental Sanitation Facilities compared to other experts.

**Table 8.** Indicators with Significant Differences in Weight Scoring Among Different Experts.

| Expert Types | C9 | D2 | D3 | D8 | D12 |
|---|---|---|---|---|---|
| District Manager | 0.075 | 0.119 | 0.375 | 0.074 | 0.205 |
| Tourism Experience Expert | 0.064 | 0.133 | 0.222 | 0.215 | 0.059 |
| Academic Expert | 0.233 | 0.625 | 0.100 | 0.275 | 0.093 |

Another area of significant difference is in factor D8 Unique Cultural Activities. Both tourism experience experts and academic experts gave this factor a higher weight, whereas district managers scored it lower. This indicates that for tourists and scholars, long-term cultural activities as a soft environmental element can provide considerable attraction. District managers explained that the occurrence and continuous operation of cultural activities are difficult to directly intervene in, and it is challenging to create influential festivals and unique events out of thin air. While successful cultural activities can be "created", such as regularly held local music festivals or anime conventions, their success does not solely come from the managers and event operators but often relies heavily on the local social and commercial environment.

Another noteworthy difference in weight is seen in D12 Market Activities. The district managers interviewed gave this factor a higher weight, whereas tourism experience experts and academic experts, as non-local residents, gave it less than half the weight of the former. This might be because market activities can bring economic benefits to the district, making them quite important in the eyes of managers. However, for non-local tourists, these activities, mainly catering to local visitors, may not be of much interest. Most market activities focus on snacks, daily necessities, and creative cultural products, with little variation in form between regions. As they mainly serve local residents, their primary content is geared towards everyday life, making it difficult to become a key factor for non-local visitors when choosing a travel destination.

*5.2. Discussion on the Multi-Source Data Method*

The results obtained through the user satisfaction questionnaire show a high consistency with the scores calculated using the multi-source data combined with AHP for itemized scoring. Table 9 reflects this itemized consistency.

**Table 9.** Comparison of Scores from Multi-Source Data Combined with AHP Method and Tourist Satisfaction Questionnaire Scores.

| Scoring Dimension | Multi-Source Data + AHP Score | Satisfaction Questionnaire Score |
|---|---|---|
| Cultural Dimension Score (C2) | PJ(0.168) > ST(0.133) > GQ(0.108) | PJ(4.200) > ST(3.850) > GQ(3.275) |
| Consumer Dimension Score (C5) | GQ(0.066) > PJ(0.055) > ST(0.041) | PJ(3.650) > GQ(3.500) > ST(3.325) |
| Tourism Dimension Score (B1) | PJ(0.557) > ST(0.419) > GQ(0.330) | PJ(4.100) > ST(3.700) > GQ(3.400) |
| Infrastructure Score (C6-C10) | PJ(0.145) > ST(0.141) > GQ(0.138) | PJ(3.925) = GQ(3.925) > ST(3.850) |

In the scoring items using multi-source data, the ratings for cultural and tourism projects in the three districts obtained the same ranking. However, in the consumer (C5) and infrastructure categories (C6–C10), a discrepancy was observed between the scores calculated from multi-source data and those from the satisfaction questionnaire. Firstly, Pingjiang district scores significantly higher in the consumer environment compared to Guanqian district. At the same time, Shantang district, which was slightly higher than Guanqian district in infrastructure scores, has now become slightly lower. This could be due to several reasons:

1.  Incompleteness of Data Sources: For example, in assessing environmental sanitation facilities, the count of public restrooms as displayed on two major Chinese mapping software platforms was used. However, restrooms located inside commercial complexes are mostly not displayed on these platforms and have led to a decrease in the scores of the commercially rich Guanqian district in the AHP evaluation.

2.  During the evaluation, scores were assigned based on simple quantity, neglecting factors such as quality and the rationality of distribution. In the evaluation process, several respondents mentioned issues in the Guanqian district, such as the aging commercial ecosystem and the homogenization of business content in many key location shops, which deteriorated their impression of the commercial environment

in the Guanqian district. However, the scoring method in the AHP evaluation, which only considered the quantity of commercial varieties, led to a significant increase in the relative score of Guanqian as a commercial district.

3.  Limitations in the Multi-Source Data Evaluation Approach: For instance, factors like transportation convenience were evaluated based on the number of subway and bus stations. However, when tourists experience local transportation facilities, they also encounter walking facilities, rest areas, etc., requiring further research to make the infrastructure evaluation more rational.

In the overall composite evaluation of the districts by tourists, Guanqian district (3.825) received the same score as Shantang district (3.825), showing a slight deviation from the AHP scoring results. Several reasons might explain this phenomenon. One reason could be the uncontrollable variables of tourists filling out the satisfaction questionnaires. Due to variations in foot traffic and weather, visiting the three historical districts at different times and dates could lead to varying experiences and, thus, affect satisfaction levels. Future research could involve increasing the number of interviewed tourists to minimize this error. Another reason might be biases in weight setting; the satisfaction survey was conducted solely with tourists, whereas experts were involved in the AHP weighting process, leading to a divergence between the weights of cultural, commercial, and tourism factors and the results of tourist evaluations. Additionally, variations in supporting factors, specifically differences in scores for indices related to consumption and infrastructure, resulted in discrepancies in the overall final scores.

*5.3. Conclusions*

This study builds an urban historical and cultural district competitiveness indicator system based on Crouch and Ritchie's theory of destination competitiveness. It calculates the weights of each indicator through expert surveys and AHP and constructs a calculation method for each indicator based on multi-source data, using Shantang, Pingjiang, and Guanqian as examples to validate the proposed system. The research results indicate that the evaluation system proposed in this paper firstly enables a more comprehensive and fine-grained consideration of tourism destinations. It transforms the abstract issue of historical district competitiveness into a tangible and quantifiable parameter system. This aids in enhancing the rationality and timeliness of interventions and measures implemented by historical district managers, such as "Should the primary focus be on the preservation of cultural resources, or on increasing basic service facilities?" and "What types of commercial activities should be introduced and developed?" Secondly, the evaluation method based on multi-source data allows for real-time and efficient implementation of evaluations, ensuring the timeliness of decision-making and better adapting to the dynamic nature of tourism destination development. However, this paper still has the following limitations:

1.  Subjectivity in the Scoring Process with Multi-Source Data: The study, while organizing and dissecting the concept of tourism competitiveness of urban historical and cultural districts, did not provide more detailed definitions for more fundamental indicators, such as "experience-oriented cultural venues" or "commercial environment". Generally, factors affecting tourist satisfaction include not only the type and quantity of services provided by the destination but also the quality of these services. Due to the complexity of the research, this paper simply used the number or type of commercial entities for scoring without delving into a more complex quality assessment.
2.  Inconsistency in Multi-Source Data Definitions: For instance, different data platforms have completely different understandings of the "Guanqian District". There is no unified standard for different data platforms to include a certain POI in a specific area, so many POIs included in the scoring might not actually be located within the area shown in Figure 5.
3.  Insufficiency in the Number of Experts and Tourist Questionnaires: In the process of AHP weighting and tourist satisfaction scoring, there is a problem of insufficient numbers of experts and tourist questionnaires.

For the first issue, future research is planned to specifically study the corresponding evaluation system and further refine it using techniques such as semantic analysis of user reviews. For instance, emotional semantic analysis technology [40] can be used in conjunction with the fuzzy comprehensive evaluation method. This approach would utilize review data from shops within the district as the data source to provide a more precise and real-time assessment of its commercial environment and service quality. For the second issue, there is a plan to develop related assessment software tools using map platform interfaces to avoid this problem. At the same time, it is necessary to increase the number of experts and survey subjects to further enhance the accuracy of the research content. To achieve this, we need to establish a long-term mechanism to amend the evaluation system. For instance, setting up an intelligent data analysis system for historical districts that can automatically collect and process data. This system could draw inspiration from the crowdsourcing model used in MIT Media Lab's Place Pulse project [41]. By inviting more experts from diverse cultural backgrounds to participate in constructing the evaluation mechanism, we aim to gather and filter tourist evaluation information from around the world, ensuring the quality and diversity of tourist feedback.

**Author Contributions:** Conceptualization, C.L., Y.L., and M.-e.H.; methodology, C.L., Y.L., and M.-e.H.; investigation, Y.L.; data analysis, Y.L., validation, C.L. and M.-e.H.; writing—original draft preparation, Y.L.; writing—review and editing, C.L. and M.-e.H.; visualization, Y.L.; funding acquisition, C.L. All authors have read and agreed to the published version of the manuscript.

**Funding:** This research was funded in part by the National Natural Science Foundation of China under Grant 52308032, in part by the Key Program of Panxi Health Care Industry Research Center of China under Grant PXKY-ZD-202304, and in part by the Program of Landscape Health Database Construction and Park Smart Management Technology of Tongji University.

**Institutional Review Board Statement:** Not applicable.

**Informed Consent Statement:** Informed consent was obtained from all subjects involved in the study.

**Data Availability Statement:** The data presented in this study are available on request from the corresponding author.

**Conflicts of Interest:** The authors declare no conflict of interest.

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
