# Peer review of "Tourism Competitiveness Evaluation Model of Urban Historical and Cultural Districts Based on Multi-Source Data and the AHP Method: A Case Study in Suzhou Ancient City"

_sustainability, doi:10.3390/su152416652_

Round 1
Reviewer 1 Report
Comments and Suggestions for Authors
This paper is of high quality, well-written, and well-crafted, demonstrating a very good grasp of technical concepts. Its structure is correct, and the content is sound. However, an important point is missing that the authors should address: the paper's relevance, often referred to as the 'so what' question. I believe the authors have failed to explain why their paper is significant. Additionally, once the relevance of the paper is presented in the Introduction, it should be briefly reinforced in the Conclusion, along with an appropriate approach to the implications of their findings for policy and intervention.
Reviewer 2 Report
Comments and Suggestions for Authors
The paper addresses the timely issue of effective and accurate assessment of tourism competitiveness in relation to historic and cultural urban districts, as they face challenges in attracting visitors due to imitating of business models - a problem particularly relevant to historic cities in China. The problem is comprehensively outlined, with appropriate reference to past and present theoretical background.
The authors rightly stress the need to consider the competitiveness of historical and cultural urban areas in the light of all their characteristics (l. 153-155).The models chosen for discussion are plausible and can provide a sound basis for assessing the tourism competitiveness of the types of urban areas in question.
However, there are several flaws in this paper that need to be addressed.
1. The paper would gain in terms of explanatory power if Table 1 ( line 255) were discussed in more detail.
2. A new sub-section should begin on line 257.
3. The tourism information platforms mentioned in lines 304-306 are not known to readers from outside China and should be better explained.
4. It is unclear why only 21 of the 25 evaluation indicators (C and D) presented in Figure 1 are discussed in Figure 2. Furthermore, some aspects of the framework presented in Figure 1 lack clarity. It is not clear what is meant by "influence of the city" (D1). In addition, aspects such as the perceived reputation of a city or any efforts by city marketing or cultural marketing departments/institutions are not included as factors in tourism competitiveness, although they play an important role in it.
5. The most problematic part of the paper begins on line 330. The authors have used questionnaires filled in by six experts representing three different categories. Then, for each category consisting of only two experts, they obtained weights/averages for each indicator discussed. The results obtained cannot be taken seriously in terms of their explanatory power, as they have no statistical significance. It is commendable that the authors consider the extremely small number of expert questionnaires analysed as a limitation of the research (lines 498-500), but the considerations based on a statistical analysis of 6 interviews lack credibility.
6. Lines 352-254 make very brief references to three historic districts of Suzhou city. It is impossible to understand from the paper what the characteristics of the selected districts are, let alone for a non-Chinese reader unfamiliar with the city, which also severely limits the explanatory power of Tables 3 and 4. Consequently, it is also impossible to follow the very brief discussion of the tourism competitiveness scores of the three districts (lines 368-372).
7. The validation of the results, carried out only on a group of 10 tourists - lines 374-378 - (it is not even specified who they were in terms of gender, age, education, etc.), further reduces the explanatory power of the results of the paper.
8. There is a typo in line 381, it should read Table 6 instead of Table 7.
In view of the above, I have no choice but to recommend a major revision of the paper.
Reviewer 3 Report
Comments and Suggestions for Authors
The paper titled "Tourism Competitiveness Evaluation Model of Urban Historical and Cultural Districts Based on Multi-Source Data and the AHP Method: A Case Study in Suzhou Ancient City" addresses the challenge of evaluating the tourism competitiveness of urban historical and cultural districts. The study proposes an evaluation method that combines multi-source data and the Analytic Hierarchy Process (AHP) to establish an evaluation framework. The framework includes dimensions such as Core Resources and Attractors, Supporting Factors and Facilities, and Guarantee Factors. The proposed model is tested on three historical districts in Suzhou, and the results are cross-validated with tourist satisfaction surveys. The findings suggest the model's effectiveness in assessing cultural, commercial, and tourism service aspects, providing a theoretical foundation for tourism decision-making systems and practical applications in historical districts.
Comments:
1) The study employs the AHP method for calculating the weights of evaluation indicators. How were the criteria for selecting and prioritizing experts in the AHP process determined, and what steps were taken to ensure the reliability and consistency of their judgments?
2) The dimensions of Core Resources and Attractors, Supporting Factors and Facilities, and Guarantee Factors were adapted from Crouch and Ritchie's model. How were these dimensions specifically tailored to the unique characteristics of urban historical and cultural districts, and were there any challenges in this adaptation process?
3) The paper discusses the subjective nature of scoring fundamental indicators like "experience-oriented cultural venues" and "commercial environment." Can you provide insights into potential methodologies or criteria that could be employed to achieve a more nuanced and objective assessment of these indicators?
4)In the section discussing the Inconsistency in Multi-Source Data Definitions, what strategies or methodologies could be employed to establish a more standardized definition of districts or points of interest across different data platforms?
5) The paper acknowledges the limitation of insufficient numbers of experts and tourist questionnaires in the AHP weighting and tourist satisfaction scoring. What strategies could be employed to increase the number of participants while maintaining the quality and diversity of expert opinions and tourist feedback?
6) The study mentions plans for future research to refine the evaluation system using techniques such as semantic analysis of user reviews. Could you elaborate on how semantic analysis could be employed, and what specific advantages or challenges might arise from this approach?
7) The research focuses on three historical districts in Suzhou. How generalizable are the findings to other urban historical and cultural districts, especially in different cultural and geographical contexts? Are there specific factors that may limit the applicability of the proposed model in certain settings?
8) The paper identifies certain limitations, such as the subjectivity in the scoring process and inconsistency in data definitions. What steps or methodologies could be implemented to mitigate these limitations in future research or practical applications?
9) The study proposes using map platform interfaces for assessment software tools. Could you discuss the potential advantages and challenges associated with implementing such tools, and how they might contribute to overcoming issues related to data inconsistency?
10)Considering the evolving nature of historical and cultural districts and the dynamic tourism market, how does the proposed model account for changes over time, and what mechanisms or updates could be integrated to ensure the model's relevance and accuracy in the long term?
Round 2
Reviewer 2 Report
Comments and Suggestions for Authors
The authors meticulously attended to all suggestions, thereby significantly enhancing the cognitive merit of the article. I harbor no additional reservations and am delighted to endorse the article for publication.
Reviewer 3 Report
Comments and Suggestions for Authors
Accept in present form.